# Online Portfolio Selection with ML Predictions

**Ziliang Zhang**
School of Computer Science
The University of Sydney
Camperdown NSW 2050, Australia
zzha0461@uni.sydney.edu.au

**Tianming Zhao**
School of Computer Science
The University of Sydney
Camperdown NSW 2050, Australia
tzha2101@uni.sydney.edu.au

**Albert Y. Zomaya**
School of Computer Science
The University of Sydney
Camperdown NSW 2050, Australia
albert.zomaya@sydney.edu.au

## Abstract

Online portfolio selection seeks to determine a sequence of allocations to maximize capital growth. Classical universal strategies asymptotically match the best constant-rebalanced portfolio but ignore potential forecasts, whereas heuristic methods often collapse when belief fails. We formalize this tension in a learning-augmented setting in which an investor observes (possibly erroneous) predictions prior to each decision moment, and we introduce the *Rebalanced Arithmetic Mean portfolio with predictions (RAM)*. Under arbitrary return sequences, we prove that RAM captures at least a constant fraction of the hindsight-optimal wealth when forecasts are perfect while still exceeding the geometric mean of the sequence even when the predictions are adversarial. Comprehensive experiments on large-scale equity data strengthen our theory, spanning both synthetic prediction streams and production-grade machine-learning models. RAM advantages over universal-portfolio variants equipped with side information across various regimes. These results demonstrate that modest predictive power can be reliably converted into tangible gains without sacrificing worst-case guarantees.

## 1 Introduction

Online portfolio selection, the sequential allocation of capital across multiple assets (e.g., stocks), sits at the crossroads of mathematical finance, statistical learning, and online algorithmic design. In every trading period, an investor must commit to a nonnegative, unit-sum weight vector before observing the next price relatives[1], relying on two imperfect guides: extracted patterns from historical returns and predictive signals on future returns that may be noisy. The puzzle is therefore two-fold: (i) distill from the past statistical regularities that remain informative, and (ii) fuse them with fallible forecasts in a way that preserves worst-case guarantees. Resolving this tension is challenging because we can neither guarantee the persistence of past regularities nor trust the accuracy of future forecasts blindly.

**Literature review.** The growth-optimal portfolio paradigm traces back to Kelly's [1] information-theoretic formulation of proportional betting, which showed that compounding wealth at the exponential rate is achievable when returns are i.i.d. Capital growth theory [2] later extended this principle to multi-asset markets, providing a rigorous benchmark for long-horizon investment under more

---

[1]Throughout, "price relative" and "return" are used interchangeably.

39th Conference on Neural Information Processing Systems (NeurIPS 2025).

realistic dynamics. Contemporary research has bifurcated into two main streams. The first is to find universal algorithms that provide provable guarantees against adversarial return sequences without committing to a return model; the second relies on heuristic assumptions or data-driven forecasting powered by machine-learning models, trading theoretical certainty for empirical performance. These complementary approaches motivate our study, we therefore begin with a overview of each.

Universal algorithms, such as Universal Portfolio [3] and Exponential Gradient Update [4], make no market assumptions about price dynamics yet remain provably competitive with the best constant-rebalanced portfolio (BCRP) selected in hindsight. After $n$ rounds, their cumulative wealth lags BCRP by at most a polynomial factor, yielding sublinear regret in logarithmic growth. However, to hedge against adversarial sequences, universal algorithms deliberately diffuse capital across assets, forfeiting the windfalls that arise when strong predictive structure is present. A more frugal member of the provably safe camp is the uniform buy-and-hold strategy [5]. Its final wealth is bounded at the arithmetic mean of single-asset returns, but the absence of rebalancing makes it even more conservative than the universal-portfolio family, and empirical evidence shows it often lags behind whenever return dispersion is substantial. Further algorithmic families are reviewed comprehensively in the survey by Li and Hoi [6]. However, despite theoretical guarantees against offline benchmarks, these traditional algorithms share a cautious bias that may overlook significant predictive gains.

On the other hand, heuristic methods ranging from mean reversion [7; 8] to pattern matching approaches [9] rely on market assumptions to inform algorithmic decisions based on signaled patterns. Furthermore, there is growing interest in employing machine-learning (ML) models to forecast future returns, heavily inclining the portfolio to the highest predicted asset in attempt to yield optimal return. Morris et al. [10] introduce an ensemble framework that combines data mining techniques with long short-term memory networks for forecasting stock and Bitcoin prices. Singh and Srivastava [11] deploy a deep neural network for stock price forecasting, demonstrating enhanced performance over recurrent neural networks. Zhang et al. [12] integrate a generative adversarial network with a long short-term memory module, achieving promising results in real-world closing-price prediction. Feng et al. [13] propose an advanced machine learning approach to stock movement prediction, employing adversarial training to enhance the generalization capability of neural networks. These prediction-based strategies can exploit favorable market conditions and achieve superior returns if the data follows the underlying model. However, when the distribution of future returns deviates from the model, these approaches may make poor predictions and suffer unbounded losses compared to more conservative methods. In addition, ML models may suffer from overfitting if trained on insufficient or unrepresentative features. As wealth compounds exponentially, even a brief sequence of poor forecasts can rapidly erode capital, leading to catastrophic losses within a short time horizon.

Navigating preservation versus prediction-driven aggressiveness brings us to a fundamental question:

*How to exploit forecasts yet preserve worst-case guarantees in portfolio selection?*

Our answer is to weave ML forecasts into the fabric of classically robust schemes. The design is guided by the emerging paradigm of algorithms with predictions [14], where an online procedure receives a potentially faulty glimpse of the future and must convert that hint into improved average-case performance while keeping any inflation of worst-case regret provably minimal. In this framework, a baseline algorithm is augmented by some predictive signals that balances follow-the-prediction against hedge-for-adversity, yielding an error-sensitive competitive ratio that smoothly interpolates between perfect-forecast optimality and classical worst-case guarantees under adversarial prediction.

The learning-augmented paradigm has been instantiated across a wide spectrum of classic problems. Lykouris and Vassilvitskii [15] demonstrate how a single tunable parameter yields an error sensitive competitive ratio for the paging problem, outperforming LRU [16] whenever the predictor is informative while retaining its worst-case bound. Kumar et al. [17] adapt the same recipe to both the ski-rental dilemma and non-clairvoyant job scheduling. Bai and Coester [18] present a sorting algorithm that harnesses potentially erroneous predictions to enhance computational efficiency. Learning-augmented techniques have also appeared in financal settings, including Pareto-optimal threshold-based algorithms for online conversion problems [19] and tight consistency-robustness bounds for One-Max-Search [20]. For a more comprehensive overview, we refer to the survey by Mitzenmacher and Vassilvitskii [21] and the online repository at [14]. Online portfolio selection shares the same sequential-decision procedure: each round demands an irrevocable allocation before the next return vector is revealed. By importing the learning-augmented toolkit from algorithms with predictions into this framework, we aim to fuse the upside of ML forecasts with the downside pro-

tection of conservative strategies, yielding wealth trajectories that improves smoothly with accurate predication while retaining a measurable offline benchmark when predictive quality degrades.

To date, very few studies have embedded modern ML forecasts directly into the update rules of classical online portfolio algorithms. The main precursor is the side-information variant of Cover's universal portfolio [22]. At every round, the investor receives a discrete signal that labels the current market regime (i.e., bull or bear) and maintains a separate universal strategy for each regime, with the goal of matching the state-dependent BCRP (BSCRP). Borodin et al. [7] instead assume a mean-reversion pattern in next-period prices, but offer no error-sensitive guarantees linking performance to the quality of its forecasts. More recent work has refined the side-information idea: Bhatt et al. [23] extend universal portfolios to continuous side information via a probabilistic partitioning that achieves first-order asymptotic optimality against BSCRP, while Yang et al. [24] employ an expert-aggregation framework to construct state-dependent expert ensembles that achieve the same asymptotic benchmark.

However, these constructions inherits several limitations. (i) Both BCRP and BSCRP are modest benchmarks, which can lag the true offline optimum that reallocates into the single best-performing asset each day by an exponential factor. (ii) Universal algorithms converge to BCRP and/or BSCRP only at a polynomial rate in wealth, so the lower bound becomes practically vacuous once the exponential gap to the global optimum is accounted for. (iii) The regret bound on growth rate with side information grows linearly with the number of states, so a richer information alphabet paradoxically weakens the guarantee. (iv) Focusing solely on asymptotic growth obscures performance over the finite horizon which matters most in practice. (v) Most critically, existing results are prediction-agnostic. They provide no quantitative characterization of how performance scales with prediction error, leaving unanswered how the algorithm fares when the side information is perfectly informative, entirely misleading, or somwehere in between.

These limitations motivate our approach, which fuses ML forecasts with adversarial safeguards and provides error-sensitive guarantees that explicitly couple portfolio performance to prediction quality. We now formalize the problem and introduce the necessary notation.

**Problem statement.** We study online portfolio selection over $n$ trading periods and $m^2$ assets. Let

$$\mathbf{x}^n = \big(\mathbf{x}(1), \mathbf{x}(2), \ldots, \mathbf{x}(n)\big), \qquad \mathbf{x}(i) = \big(x_1(i), x_2(i), \ldots, x_m(i)\big) \in \mathbb{R}_+^m \tag{1}$$

where $x_j(i)$ is the price relative of asset $j$ from period $i-1$ to $i$. A portfolio strategy defines

$$\mathbf{b}^n = \big(\mathbf{b}(1), \mathbf{b}(2), \ldots, \mathbf{b}(n)\big), \qquad \mathbf{b}(i) = \big(b_1(i), \ldots, b_m(i)\big) \in \Delta^{m-1}$$

with $b_j(i) \geq 0$ and $\sum_{j=1}^m b_j(i) = 1$ (self-financed, no margin/shorting). The wealth after $n$ periods is

$$S_n\big(\mathbf{b}^n, \mathbf{x}^n\big) = \prod_{i=1}^n \big(\mathbf{b}(i) \cdot \mathbf{x}(i)\big) = \prod_{i=1}^n \sum_{j=1}^m b_j(i) \, x_j(i)$$

We set the initial capital to $S_0 = 1$; thus, $S_n$ equals both the terminal wealth and total growth factor. At the start of each period $i$, the investor receives a ranking forecast from a black-box ML oracle. Let $[m] := \{1, \ldots, m\}$. The oracle outputs a permutation $\sigma(i)$ of $[m]$, where $\sigma(i)_k$ denotes the index of the asset ranked $k$-th (highest) by predicted one-step return. For analysis, we define $\mathbf{y}(i)$ by reindexing the realized returns according to the predicted ranking:

$$\mathbf{y}(i) := \big(y_1(i), \ldots, y_m(i)\big), \qquad y_k(i) := x_{\sigma(i)_k}(i) \tag{2}$$

Thus $y_1(i) \geq y_2(i) \geq \cdots \geq y_m(i)$ reflects the oracle's predicted order (highest to lowest). Importantly, our algorithm and guarantees depend only on the ranking $\sigma(i)$; we do not assume access to, nor require, predicted magnitudes of next-period returns. The vector $\mathbf{y}(i)$ is introduced solely as analysis notation to track the realization under the predicted order.

We make no assumptions about how this vector (ranking) is generated, and it may come from any ML model, e.g.: a neural network [10] that predicts one-step-ahead returns and then ranks them; a gradient-boosted tree model [25] that outputs the ranking directly; or any heuristic ranking rules. For analysis, we also define the clairvoyant order statistics $x_{(1)}(i) \geq \cdots \geq x_{(m)}(i)$ as the entries of $\mathbf{x}(i)$ sorted in

---

[2]We fix the number of assets $m$ at the outset and treat it as a constant.

non-increasing order, and write $\mathbf{x}^{\downarrow}(i) \coloneqq (x_{(1)}(i), \ldots, x_{(m)}(i))$ and $\mathbf{x}^{\uparrow}(i) \coloneqq (x_{(m)}(i), \ldots, x_{(1)}(i))$. Note the distinction: $x_j(i)$ is the return of asset $j$ (original index), whereas $x_{(j)}(i)$ is the $j$-th largest return at time $i$ (ranked value). Ties, when present, are broken by a fixed deterministic rule; following analysis are invariant to this choice. We apply the same shorthand to the current weights $\mathbf{b}^{\downarrow}(i)$:

$$\mathbf{b}^{\downarrow}(i) = \big(b_{(1)}(i), b_{(2)}(i), \cdots, b_{(m)}(i)\big) \tag{3}$$

Our goal is to design an online portfolio algorithm with prediction that translates the oracle's noisy forecasts $\mathbf{y}^n$ into higher wealth whenever they are informative, and preserves a provable worst-case guarantee matching the best prediction-agnostic baseline when the forecasts are arbitrarily inaccurate. We adopt the geometric mean of all returns referring the *value line index*, as such a baseline. Formally,

$$S_n^{\text{GM}} = \left[ \prod_{j=1}^{m} \prod_{i=1}^{n} x_j(i) \right]^{1/m} \tag{4}$$

The best-known competitive [26] prediction-agnostic strategy is the Universal Portfolio [3], which only guarantees wealth no less than $S_n^{\text{GM}}$ [3, Prop. 5]. Matching this floor therefore secures state-of-the-art worst-case performance while leaving headroom for upside when the oracle proves informative.

**Our contribution.** We propose a learning-augmented portfolio with updating rules guided by ML forecasts, the Rebalanced Arithmetic Mean (RAM). For the prediction error[3] $\eta_n \in (0, 1]$ induced by any predictions $\mathbf{y}^n = (\mathbf{y}(1), \mathbf{y}(2), \cdots, \mathbf{y}(n))$, the resulting wealth of RAM after $n$ periods satisfies:

$$S_n^{\text{RAM}} = \prod_{i=1}^{n} \mathbf{b}^{\downarrow}(i) \cdot \mathbf{y}(i) \geq \max \left\{ S_n^{\text{GM}} = \left[ \prod_{j=1}^{m} \prod_{i=1}^{n} x_j(i) \right]^{1/m}, \ \eta_n \prod_{i=1}^{n} \mathbf{b}^{\downarrow}(i) \cdot \mathbf{x}^{\downarrow}(i) \right\}$$

with weight updating rules $\mathbf{b}^{\downarrow}(i)$ followed by rank-matching strategy[4] using predicted permutations:

$$\mathbf{b}^{\downarrow}(i) = \big(b_{(1)}(i), \cdots, b_{(m)}(i)\big), \quad b_k(i) = \frac{b_{(j)}(i-1)y_j(i-1)}{\sum_{j=1}^{m} b_{(j)}(i-1)y_j(i-1)}, \quad b_k(1) = \frac{1}{m}, \forall i, j, k \tag{5}$$

When the predictions are maximally informative $\eta_n = 1$ and $\mathbf{y}(i) = \mathbf{x}^{\downarrow}(i), \forall i$, RAM achieves wealth at least a constant fraction of the hindsight-optimal "all-in-best-asset" benchmark:

$$S^{\text{RAM}} \geq \frac{1}{m} S_n^{\text{OPT}} \coloneqq \frac{1}{m} \prod_{i=1}^{n} \max_j x_j(i)$$

Conversely, when forecasts are actively hostile (adversarial), we prove that RAM dominates the value line $S_n^{\text{GM}}$. The results are independent of market assumptions other than positive return factors.

We substantiate our theory with real-world equity experiments in progressively richer settings:

1. Prediction-free benchmark: We show that a randomized, prediction-free variant of RAM outperforms the Universal Portfolio in expectation.

2. Controlled-noise study: Using synthesized predictions under controlled accuracy thresholds, RAM degrades gracefully under heavy noise and raises smoothly with improved predictions.

3. Real-world deployment: We deploy RAM with real-world ML model on recent market data, achieving significant gains over existing baselines, underscoring its practical utility.

## 2 Portfolio with predictions

**Blindly following the forecasts.** Consider two assets $x_1$ and $x_2$, where one either doubles or halves while the other does the opposite. Let the forecast be wrong with independent probability $\epsilon \in (0, 1)$. An investor who completely follows the forecast accumulates wealth $S_n = 2^{(1-\epsilon)n}(0.5)^{\epsilon n} = 2^{(1-2\epsilon)n}$. When $\epsilon < 0.5$, the investor still gains, but at an exponential gap of $2^{2\epsilon n}$ against the optimum $S_n^* = 2^n$. Conversely, when $\epsilon > 0.5$, the wealth decays exponentially to zero. Therefore, even a small error rate produces unbounded competitive loss and expose the strategy to total ruin.

---

[3]Formally defined in Section 2.1 (Eq. 6)

[4]Detailed in Section 2.1; see Algorithm 1 for pseudocode implementation.

**Partially trusting the forecast.** A natural repair is to invest a fraction $\lambda \in [0, 1]$ in the predicted winner and keep the remaining in a safe baseline such as the uniform portfolio $(0.5, 0.5)$. Lemma A.1 (Appendix) proves that, when the forecast is wrong with independent probability $\epsilon \in (0, 1)$, the expected wealth is $\mathbb{E}[S_n] = 1.25^n(1 + 0.6\lambda(1 - 2\epsilon))^n$. Consequently, the expected per-period log-growth is $W_n(\lambda, \epsilon) = \log 1.25 + (1 - \epsilon) \log (1 + 0.6\lambda) + \epsilon \log (1 - 0.6\lambda)$. For every fixed $\lambda \geq 1/3$, there always exists a $\epsilon^*(\lambda) < 0.5$ forcing $W_n(\lambda, \epsilon^*) = 0$. If an adversary pushes the error rate even slightly above this threshold, $W_n(\lambda, \epsilon) < 0$ and wealth decays exponentially to zero. Meanwhile the competitive ratio against the optimal hindsight $2^n$ still diverges exponentially whenever $\lambda > 0$ and $\epsilon > 0$. Thus, fractional betting ameliorates but does not eliminate worst-case fragility.

Our intuition is to decouple magnitudes from directional signs through a two-stage architecture:

1. *Base generator.* At each period, a prediction-agnostic strategy updates a weight vector using only realized return history, preserving the base algorithm's worst-case guarantees.

2. *Permutation layer.* The ML predicted ranking (the ordering of upcoming returns) then re-labels these weights, concentrating larger masses on assets it considers promising.

This design inherits the robustness of the base generator yet exploits predictive insights whenever it is present. The crux is to pair a provably safe generator with a permutation rule that minimally degrades its guarantee while delivering gains under accurate predictions. The remainder of the paper therefore concentrates on this permutation-based portfolio class.

## 2.1 Rebalanced arithmetic mean

We propose the Rebalanced Arithmetic Mean (RAM) portfolio which overlays an oracle-supplied ranking of next-period gross returns onto the buy-and-hold baseline. Let the baseline assign weights $\{b_j(i)\}_{j=1}^m$ at period $i$ according to equation 5, where the pipeline follows by matching higher weights to higher predicted returns according to their ranks. Concurrently, an oracle outputs a permutation $\sigma(i)$ intended to rank the forthcoming returns $\mathbf{x}(i)$; applying it produces the oracle-ordered vector $\mathbf{y}(i)$ as defined in equation 2. Conceptually, RAM is still a buy-and-hold policy, yet executed in a dynamically permuted coordinate system determined by permutations $\{\sigma(i)\}_{i=1}^n$.

RAM operates by greedy matching: it pairs the largest weight $b_{(1)}(i)$ with the asset which the oracle predicts to deliver the highest return, the second-largest weight with the second-highest predicted return, and so on. The portfolio held at period $i$ therefore earns the inner product $\mathbf{b}^{\downarrow}(i) \cdot \mathbf{y}(i)$. Importantly, we highlight the key relation of $\mathbf{b}^{\text{new}} \cdot \mathbf{x}(i) = \mathbf{b}^{\downarrow}(i) \cdot \mathbf{y}(i)$ as in Algorithm 1. A visual example of RAM's update process is shown in Figure 1.

Figure 1: RAM's rank-matching architecture: larger weights paired with higher predicted ranks.

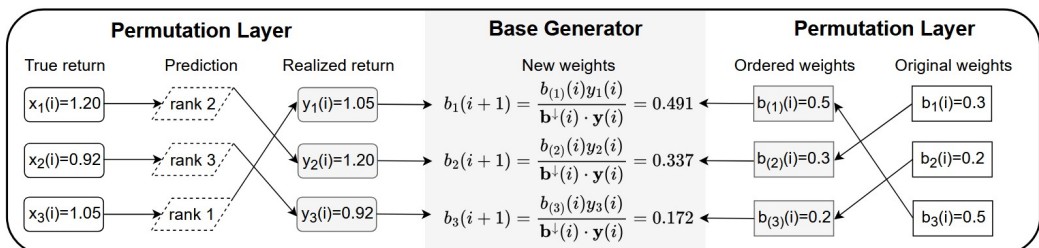

To quantify the oracle's accuracy, we compare its ordering with the clairvoyant optimal ordering of the upcoming returns $\mathbf{x}^{\downarrow}(i)$. We define the per-period error rate $\eta(i)$ and total error rate[5] $\eta_n$:

$$\eta(i) := \frac{\mathbf{b}^{\downarrow}(i) \cdot \mathbf{x}^{\downarrow}(i)}{\mathbf{b}^{\downarrow}(i) \cdot \mathbf{y}(i)} = \frac{\sum_{j=1}^m b_{(j)}(i) x_{(j)}(i)}{\sum_{j=1}^m b_{(j)}(i) y_j(i)}, \qquad \eta_n := \prod_{i=1}^n \frac{1}{\eta(i)} \qquad (6)$$

---

[5]The daily error rate $\eta(i)$ is defined so that it decreases as predictions improve and increases as they degrade. The aggregated quantity $\eta_n$ behaves oppositely; although we phrase it as an overall error rate for brevity, it should be interpreted as a performance metric, where larger values indicate better performance.

**Algorithm 1** RAM: rebalanced arithmetic mean with predictions

---

**Require:** return matrix $\{\mathbf{x}(i)\}_{i=1}^{n} \subset \mathbb{R}_{+}^{m}$, oracle permutation $\sigma(i) \leftarrow \textsc{ForecastPerm}(\mathbf{x}(i))$
 1: **initialize** weights $\mathbf{b}(1) \leftarrow (\frac{1}{m}, \ldots, \frac{1}{m})$, wealth $S(0) \leftarrow 1$
 2: **for** $i \leftarrow 1$ **to** $n$ **do**
 3:     Receive a ranking prediction $\sigma(i)$ from oracle.
 4:     $\mathbf{b}^{\downarrow} \leftarrow \text{sort}^{\downarrow}(\mathbf{b}(i))$
 5:     $\mathbf{b}^{\text{new}} \leftarrow \mathbf{0}$
 6:     **for** $j \leftarrow 1$ **to** $m$ **do**                                        ▷ reverse pairing
 7:         $\mathbf{b}^{\text{new}}_{\sigma(i)_j} \leftarrow \mathbf{b}^{\downarrow}_j$
 8:     **end for**
 9:     Receive return $\mathbf{x}(i)$.
10:     $R(i) \leftarrow \mathbf{b}^{\text{new}} \cdot \mathbf{x}(i)$
11:     $S(i) \leftarrow S(i-1) \times R(i)$
12:     $\mathbf{b}(i+1) \leftarrow \big(\mathbf{b}^{\text{new}} \odot \mathbf{x}(i)\big) / R(i)$
13: **end for**
14: **return** $S(n)$

---

The cumulative penalty $\eta_n$ captures the aggregate loss in wealth relative to the clairvoyant optimal matcher. The final aggregated wealth of RAM can therefore be expressed in exchangeable forms:

$$S_n^{\text{RAM}} = \prod_{i=1}^{n} \mathbf{b}^{\downarrow}(i) \cdot \mathbf{y}(i) = \prod_{i=1}^{n} \frac{\mathbf{b}^{\downarrow}(i) \cdot \mathbf{x}^{\downarrow}(i)}{\eta(i)} = \eta_n \prod_{i=1}^{n} \sum_{j=1}^{m} b_{(j)}(i) x_{(j)}(i) \qquad (7)$$

This multiplicative decomposition cleanly separates the contribution of the underlying buy-and-hold strategy from the oracle-induced error factor. When the oracle's ranking is accurate, $\eta_n = 1$ and RAM achieves the clairvoyant optimum. Even under maximally adversarial rankings, $\eta_n \in (0, 1]$, RAM's wealth cannot drop below the value line index (the lower envelope of the arithmetic mean).

**The reason RAM works.**  RAM is built on two simple intuitions. The first design goal is the immediate optimality under perfect forecast. Universal portfolios with side information [22; 4] only promise to approach the optimum in the limit, which means that they can lag for thousands of rounds and still be theoretically "optimal". RAM avoids that delay: a perfect oracle yields the exact clairvoyant growth for each and every period. The second goal is to prove disciplined loss guarantee under adversarial regime. As inspired by the arithmetic-geometric mean (AM-GM) inequality, we note a key invariance: a uniform buy-and-hold portfolio is unaffected by any fixed permutation of asset coordinates. Once the ordering is frozen, the AM-GM lower envelope applies to the entire return sequence, independent of the chosen coordinate system. Consequently, re-indexing the next-period returns is essentially cost-free, and the resulting wealth trajectory remains above the value line. Theorem 2.1 formalizes our claim with tight lower bounds under different levels of prediction quality.

**Theorem 2.1.** *For arbitrary market sequence* $\mathbf{x}^n$ *as specified in equation 1 and predictions* $\mathbf{y}^n = (\mathbf{y}(1), \mathbf{y}(2), \cdots, \mathbf{y}(n))$ *where each* $\mathbf{y}(i)$ *follows by equation 2, we have claims for Algorithm 1:*

  *1. For arbitrary predictions (worst-case guarantee):*

$$S_n^{RAM} \geq \left[ \prod_{j=1}^{m} \prod_{i=1}^{n} x_j(i) \right]^{1/m}, \quad \forall \eta_n \in (0, 1] \qquad (8)$$

  *2. Under perfect predictions (best-case guarantee):*

$$S_n^{RAM} \geq \frac{1}{m} S_n^{OPT} = \frac{1}{m} \prod_{i=1}^{n} \max_{j} x_j(i), \qquad for\ \eta_n = 1 \qquad (9)$$

*Proof.* We first prove 9. When predictions are perfect, $\mathbf{y}(i) = \mathbf{x}^{\downarrow}(i)\ \forall i$ and $\eta_n = 1$:

$$S_n^{\text{RAM}} = \prod_{i=1}^{n} \mathbf{b}^{\downarrow}(i) \cdot \mathbf{x}^{\downarrow}(i) = \prod_{i=1}^{n} \sum_{j=1}^{m} b_{(j)}(i) x_{(j)}(i)$$

By the monotonicity of $\mathbf{b}^\downarrow(i)$ and $\mathbf{x}^\downarrow(i)$:

$$b_{(1)}(i) \geq \cdots \geq b_{(m)}(i) \text{ and } x_{(1)}(i) \geq \cdots \geq x_{(m)}(i) \implies b_{(1)}(i)x_{(1)}(i) \geq \cdots \geq b_{(m)}(i)x_{(m)}(i)$$

Therefore, the index carrying larger weight at period $i$ must also carry larger wealth share after $i$:

$$b_k(i+1) = \frac{b_{(j)}(i)x_{(j)}(i)}{\sum_{j=1}^m b_{(j)}(i)x_{(j)}(i)} \implies b_{(j)}(i+1) = \frac{b_{(j)}(i)x_{(j)}(i)}{\sum_{j=1}^m b_{(j)}(i)x_{(j)}(i)}$$

Apply $b_{(j)}(n+1)$ and expand the right-hand side recursively down to period 1 yields:

$$b_{(j)}(n+1) = \frac{1}{m}\frac{\prod_{i=1}^n x_{(j)}(i)}{\prod_{i=1}^n \sum_{j=1}^m b_{(j)}(i)x_{(j)}(i)} \implies \sum_{j=1}^m b_{(j)}(n+1) =$$

$$\frac{1}{m}\frac{\sum_{j=1}^m \prod_{i=1}^n x_{(j)}(i)}{\prod_{i=1}^n \sum_{j=1}^m b_{(j)}(i)x_{(j)}(i)} = 1 \implies S_n^{\text{RAM}} = \frac{1}{m}\sum_{j=1}^m \prod_{i=1}^n x_{(j)}(i) \geq \frac{1}{m}\prod_{i=1}^n x_{(1)}(i) = \frac{1}{m}S_n^{\text{OPT}}$$

proving 9. Next, we prove 8. Recall under adversarial predictions $S_n^{\text{RAM}} = \prod_{i=1}^n \mathbf{b}^\downarrow(i)\mathbf{x}^\uparrow(i)$ and:

$$b_k(i+1) = \frac{b_{(j)}(i)x_{(m-j+1)}(i)}{\sum_{j=1}^m b_{(j)}(i)x_{(m-j+1)}(i)} \implies b_k(i+1) = \frac{b_k(i)x_{\overline{\sigma}(i)_k}(i)}{\sum_{j=1}^m b_k(i)x_{\overline{\sigma}(i)_k}(i)}$$

where we have re-indexed the assets relative to the weight vector such that $\overline{\sigma}(i)_k$ denotes the adversarially permuted pairing on weight $b_k(i)$, so the algebra is carried out in a moving coordinate system attached to the weights, not in the fixed asset labeling. The specific indices $\overline{\sigma}(i)_k$ are irrelevant to the wealth analysis, since the only property we need is that every return value $x_{\overline{\sigma}(i)_k}(i)$ is paired with exactly one weight coordinate. Apply $b_k(n+1)$ and expand the right-hand side yields:

$$b_k(n+1) = \frac{1}{m}\frac{\prod_{i=1}^n x_{\overline{\sigma}(i)_k}(i)}{\prod_{i=1}^n \mathbf{b}^\downarrow(i)\mathbf{x}^\uparrow(i)} \implies S_n^{\text{RAM}} = \frac{1}{m}\sum_{k=1}^m \prod_{i=1}^n x_{\overline{\sigma}(i)_k}(i) \geq \left[\prod_{k=1}^m \prod_{i=1}^n x_k(i)\right]^{1/m}$$

where the right-hand side follows by the arithmetic-geometric mean inequality. $\square$

Relative to existing universal strategies, RAM occupies a distinctive midpoint. Under a perfect oracle, RAM extracts a constant $1/m$ fraction of the hindsight optimum, whereas the side-information universal portfolio family converges only to the modest best state-constant rebalanced portfolio and does so asymptotically. At the opposite end of the information spectrum, both RAM and universal portfolios enjoy the same distribution-free safe net, the value line index, ensuring no catastrophic under-performance. The gap between these extremes is where RAM's permutation layer becomes pivotal. Because weights are reassigned wholesale according to the forecast ordering, RAM reacts sharply on informative ranking, but with higher variance when the signal is noisy. Recent advances in machine learning have begun to yield predictors with demonstrably non-trivial forecasting power. RAM can harness these models immediately while its AM-GM floor caps downside risk. In addition, the algorithm runs in $O(m \log m)$ per round, making it readily deployable on large-scale portfolios where the exponential costs of richer universal mixtures are prohibitive.

Table 1: Stock combinations on NYSE.

| NYSE(O) | NYSE(N) |
|---|---|
| 1: Iro & Kin | 7: Cok & Ahp & Gm & Ge & Sc |
| 2: Com & Kin | 8: Dow & Mmm & Ame & Mer & Jnj |
| 3: Com & Mei | 9: Ame & Kim & Alc & Ahp & Mor & Cok & Pan & Gm |
| 4: Luk & Kin & Arc | 10: Mmm & Dow & Mer & Ge & Ibm & Ing & Hp & Jnj |
| 5: Shr & Ge & Kin | 11: Ge & Mer & Gm & Ahp & Jnj & Pan & Ame & Ing & For & Dup & Dow |
| 6: Gul & Esp & Pil | 12: Alc & Cok & Hp & Ibm & Kim & Mmm & Mor & Sc & Gm & Ahp & Pan |

# 3 Empirical study

We begin by assessing RAM on the canonical New York Stock Exchange (NYSE) benchmark, using i.i.d. random rankings to model a fully oblivious and uninformative oracle. Our primary dataset is the original NYSE(O) collection [27], which contains 36 stocks spanning 22 years (1962–1984) over 5,651 trading days. To capture a broader range of market volatility and ensure more recent coverage, we also consider the extended NYSE(N) dataset [28], encompassing 21 assets from 1962 to 2006 (11,178 trading days). To avoid duplication, we preprocess NYSE(N) by retaining only the period from 1985 to 2006 (5,526 trading days), thus removing any overlap with NYSE(O). Each data point represents the ratio of an asset's price on a given trading day to its price on the previous day. Table 1 details the various combinations of stocks: Combinations 1–3 have been widely adopted in seminal works such as [3; 4], Combinations 4–6 follow those introduced by Yang et al. [29], and the remaining combinations were chosen for additional diversity and expanded market dimension.

To study RAM's behavior under controllable signal quality, we draw rank-predictions from a one-parameter family $\{D_p\}_{p \in [0,1]}$ which interpolates linearly between adversarial and optimal orderings:

$$
\mathcal{D}_p = \begin{cases} (1-2p)\,\delta_{\sigma^\uparrow} \ + \ 2p\,U, & 0 \le p \le \frac{1}{2}, \\ 2(1-p)\,U \ + \ (2p-1)\,\delta_{\sigma^\downarrow}, & \frac{1}{2} < p \le 1. \end{cases} \tag{10}
$$

where $\sigma^\uparrow$ and $\sigma^\downarrow$ are the clairvoyant ascending and descending orderings of the upcoming returns, $U$ is the uniform distribution over all $m!$ permutations, and $\delta_\sigma$ denotes a Dirac mass at permutation $\sigma$. Consequently, $p = 0$ produces the worst-case forecast (deterministically ascending), $p = \frac{1}{2}$ gives full randomness, and $p = 1$ provides the best-case forecast (deterministically descending). Moving $p$ away from the middle point linearly transfers probability mass from the uniform component to the appropriate Dirac measure, facilitating a transparent and smoothly tunable notion of forecast quality. For each fixed $p$ we draw one permutation per period: with probability equal to the Dirac weight we output the deterministic order; otherwise we apply a Fisher-Yates shuffle. The identically sampled permutation is fed to every applicable strategy under comparison, ensuring that performance differences arise solely from algorithmic design rather than additional stochasticity.

For each accuracy threshold $p$, we run 1,000 independent Monte Carlo simulations to obtain stable performance estimates. Table 2 summarizes the mean, standard deviation, and median, thereby reducing sensitivity to distributional skewness. For combinations 1 to 6, we ablate our method against the Universal Portfolio (UP) [3] and its side-information variant [22], owning to their strong theoretical guarantees. Because the exact UP incurs exponential computational cost in the number of assets, we also benchmark the Exponential Gradient (EG) [4] universal portfolio and its side-information extension on combination 7 to 12, which scale more favorably to large-scale portfolios. To ensure a fair comparison with side-information portfolios, we recast each forecast as a categorical signal: the asset assigned rank 1 defines the current state, yielding $m$ possible states. Under perfect forecasts, the resulting best state-constant rebalanced portfolio (BSCRP) matches the hindsight optimum. Conversely, the ranking that is adversarial to RAM still offers SI-portfolios exploitable structure, since every state inevitably identifies at least one low-return asset. Constructing a ranking that is simultaneously worst-case for both methods is impractical, we therefore adopt this mapping, recognizing that it is conservative for RAM. All experiments compute under 6h on one standard CPU. Source code is available at `https://github.com/mroymd/OPML`.

Our controlled Monte Carlo study reveals a clear, monotone relationship between forecast quality and portfolio performance. When the oracle is completely uninformative (RAM-R), our algorithm still outperforms the vanilla UP in expectation, albeit with higher variance; its median wealth remains at least twice that of the geometric-mean benchmark. On the other hand, UPSI exhibits lower dispersion and reliably tracks UP, but its expected return trails RAM-R. Introducing even a modest informational edge at 53% forecast accuracy dramatically shifts the landscape: RAM now dominates both EGSI and standard EG in both mean and median wealth. The advantage widens with portfolio dimension: the regret for EGSI grows exponentially in the number

Figure 2: Comb. 1: RAM vs. EGSI over $p \in [0.5, 0.6]$. Similar trends hold for other comb. and $p \in (0.6, 1]$.

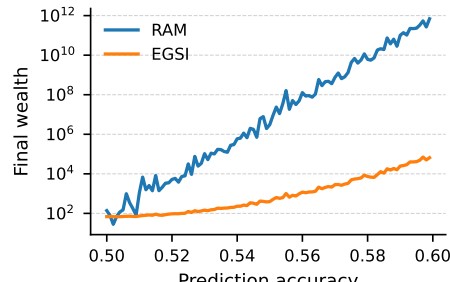

Table 2: Synthetic-forecast ablation on the NYSE dataset. Subscripts denote the forecast-accuracy parameter $p$. Reported statistics comprise the mean ($\mu$), standard deviation ($\sigma$), and median ($\overline{\mu}$). Results average over 1,000 Monte Carlo trials, each drawing predictions according to equation 10.

| Comb. | GM | UP | RAM-R $p=50\%$ | | | UPSI $p=50\%$ | | |
|---|---|---|---|---|---|---|---|---|
| | | | $\mu$ | $\sigma$ | $\overline{\mu}$ | $\mu$ | $\sigma$ | $\overline{\mu}$ |
| 1 | 6.06 | 39.97 | 57.78 | 215.75 | 13.28 | 52.33 | 36.81 | 40.57 |
| 2 | 14.65 | 80.53 | 136.39 | 551.33 | 32.15 | 97.32 | 64.84 | 79.14 |
| 3 | 34.52 | 74.07 | 101.03 | 185.44 | 53.84 | 84.88 | 22.54 | 77.26 |
| 4 | 6.69 | 32.18 | 45.66 | 107.83 | 17.81 | 40.61 | 10.68 | 37.13 |
| 5 | 5.96 | 24.40 | 35.49 | 96.26 | 15.14 | 30.74 | 8.58 | 27.95 |
| 6 | 19.35 | 47.11 | 56.97 | 73.23 | 36.53 | 55.94 | 9.84 | 53.23 |

| Comb. | GM | EG | RAM $p=53\%$ | | | EGSI $p=53\%$ | | |
|---|---|---|---|---|---|---|---|---|
| | | | $\mu$ | $\sigma$ | $\overline{\mu}$ | $\mu$ | $\sigma$ | $\overline{\mu}$ |
| 7 | 12.88 | 21.74 | 1.5E+03 | 1.9E+03 | 9.3E+02 | 22.94 | 0.43 | 22.89 |
| 8 | 22.85 | 33.37 | 1.0E+03 | 9.3E+02 | 7.2E+02 | 34.80 | 0.44 | 34.77 |
| 9 | 17.61 | 30.18 | 4.5E+03 | 6.3E+03 | 2.6E+03 | 31.18 | 0.29 | 31.15 |
| 10 | 18.43 | 31.50 | 4.6E+03 | 6.5E+03 | 2.6E+03 | 32.82 | 0.36 | 32.78 |
| 11 | 16.59 | 27.81 | 6.1E+03 | 8.2E+03 | 3.5E+03 | 28.61 | 0.17 | 28.60 |
| 12 | 14.67 | 27.86 | 1.5E+04 | 2.3E+04 | 7.3E+03 | 28.82 | 0.22 | 28.81 |

| Comb. | RAM $p=100\%$ | EGSI $p=100\%$ | RAM $p=60\%$ | | | EGSI $p=60\%$ | | |
|---|---|---|---|---|---|---|---|---|
| | | | $\mu$ | $\sigma$ | $\overline{\mu}$ | $\mu$ | $\sigma$ | $\overline{\mu}$ |
| 7 | 1.1E+41 | 4.9E+07 | 6.3E+08 | 8.6E+08 | 3.4E+08 | 36.25 | 2.35 | 36.19 |
| 8 | 1.5E+35 | 6.4E+05 | 5.8E+07 | 7.0E+07 | 3.8E+07 | 47.76 | 2.05 | 47.70 |
| 9 | 1.3E+49 | 6.7E+04 | 2.8E+10 | 4.6E+10 | 1.3E+10 | 39.59 | 1.15 | 39.52 |
| 10 | 3.4E+48 | 1.4E+05 | 2.0E+10 | 3.4E+10 | 1.0E+10 | 42.47 | 1.43 | 42.34 |
| 11 | 2.5E+53 | 2.9E+03 | 1.5E+11 | 2.3E+11 | 7.3E+10 | 33.18 | 0.59 | 33.14 |
| 12 | 2.3E+59 | 1.8E+04 | 2.5E+12 | 5.0E+12 | 1.1E+12 | 35.10 | 0.83 | 35.08 |

of states, whereas RAM's guarantees are dimension-independent, yielding superior performance in high-asset markets. At $60\%$ accuracy, the gap becomes exponential. Under perfect forecasts where BSCRP coincides with the hindsight optimum: EGSI still converges only asymptotically, whereas RAM immediately captures the optimal growth. These results show that RAM exploits accurate predictive signals more effectively than universal-portfolio baselines; Figure 2 illustrates the resulting monotone gap. We verified the sub-$50\%$ accuracy regime but omit the plots for brevity: the trajectories closely match our theoretical bound, with RAM's wealth smoothly converging to the value line without ever crossing below yet retaining domination in expectation. Collectively, these results demonstrate that RAM amplifies even modest predictive edges while provably guarding against catastrophic drawdowns.

We further evaluate RAM on a live, production-grade setting that couples real market data with an industrial-strength learning-to-rank engine. We use the nightly-refreshed S&P 500 historical panel [30] available on Kaggle, containing 501 constituents from 2010 to 2024. The index is liquid, broad-based, and widely adopted in empirical-finance research, ensuring reproducibility and practical relevance. To stress-test robustness, we analyze two disjoint windows: (i) COVID-19 crash from July 2019 to May 2020, capturing extreme volatility; (ii) Recent market from Dec 2022 to Dec 2024, reflecting contemporary trading conditions. We construct three baskets as in Table 3: (i) $\alpha$-Popular: High-capitalization names favored by retail brokers, mirroring real-world deployability; (ii) $\beta$-Diversified: Two tickers from each of five GICS sectors for cross-sector hedging; (iii) $\gamma$-Large-scale: A broad slice of the index to probe scalability. Full stock names are in Table 5.

We employ LightGBM LambdaMART [25] to forecast the ranks of next-day returns. The model is retrained each trading day using a 250-day sliding window, featuring contemporaneous and three lagged returns per asset. A decaying factor with $\theta = 0.995^{\text{age}}$ prioritizes recent observations while discarding stale information. A 60-day hold-out slice inside the same window provides early-stopping signals, eliminating look-ahead bias. This protocol guarantees strict online deployment where only data available at decision time enter either training or validation. Hyper-parameters and code are

Table 3: Basket specifications for the S&P 500 experiments. Regime A spans July 2019 to May 2020; regime B covers December 2022 to December 2024.

| Market Regime | Comb. $\alpha$ | Comb. $\beta$ | Comb. $\gamma$ |
|---|---|---|---|
| A: High volatility | Popularity-weighted | Diversified sector | Random draw |
| B: Recent window | 5 stocks | 10 stocks | 30 stocks |

supplied in the supplementary material while we keep the exposition concise to focus on RAM's prediction-integrated yet risk-free contribution.

Table 4: Empirical results on S&P 500 with production-grade machine-learning forecasts. *Best* denotes the clairvoyant benchmark of the single top-performing stock. *ML* allocates all capital each round to the asset with the highest predicted return (rank-1) from the model.

| Market Regime | Comp. | Best | GM | RAM | EGSI | ML |
|---|---|---|---|---|---|---|
| A (Covid-crash) | $\alpha$ | 1.733 | 1.206 | 1.239 | 1.230 | 1.305 |
| | $\beta$ | 1.358 | 0.953 | 1.003 | 0.982 | 1.678 |
| | $\gamma$ | 1.733 | 0.885 | 0.915 | 0.924 | 0.532 |
| B (Recent-window) | $\alpha$ | 8.293 | 2.470 | 2.512 | 2.629 | 1.676 |
| | $\beta$ | 3.153 | 1.300 | 1.377 | 1.373 | 0.572 |
| | $\gamma$ | 8.293 | 1.401 | 1.493 | 1.501 | 1.391 |

Across both market regimes, RAM consistently outperforms EGSI whenever the forecast surpass the value-line benchmark and remains resilient even when prediction collapses. In the turbulent COVID-19 window (regime A): RAM systematically exploits informative forecasts, capturing an additional 10.9% and 2.6% of excessive predictive edge over EGSI in baskets $\alpha$ and $\beta$ respectively; when forecasts turn catastrophic in basket $\gamma$, RAM stays within $1\%$ of EGSI while still eclipsing the value line by 3.3%. In the recency-focused window (regime B), the gap widens: although every forecast underperforms the value line, RAM trails EGSI by only 4.5% in basket $\alpha$ and 0.6% in basket $\gamma$. Strikingly, under the worst-case basket $\beta$ where the machine-learning signal lags the value line by 56%, RAM nevertheless beats EGSI by 2.9% and outstrips the value line by 5.9%. In aggregate, these findings provide compelling empirical evidence that RAM converts even modest predictive cues into tangible gains while preserving a universal-style safety net when those cues deteriorate. Notably, this robustness is achieved with an off-the-shelf rank predictor trained solely on historical returns within a narrow asset universe, which suggests that more sophisticated, finance-driven models would amplify RAM's advantage even further, as corroborated by our synthetic simulations.

# 4  Conclusion

This work advances learning-augmented portfolio selection by introducing the Rebalanced Arithmetic Mean portfolio with predictions (RAM), a principled framework that overlays machine-learning forecasts on a classical rebalancing rule. We prove that RAM captures a constant fraction of the hindsight-optimal wealth when forecasts are perfect and dominates the market's geometric mean baseline even under maximally adversarial signals. Extensive experiments on real-world equity data complements the theory, spanning both synthetic forecast simulations and real-world ML models: RAM harnesses informative predictive signs more effectively than side-information universal portfolios, delivering superior risk-adjusted returns across turbulent and benign regimes. These findings evidence the practical use of forecast without sacrificing worst-case safety

An important practical dimension left unexplored is transaction cost. Because RAM rebalances every round through a permutation layer, it incurs trading frictions analogous to those faced by universal portfolios. The attendant drag on portfolio wealth merits systematic investigation, for instance, through transaction-cost frameworks exemplified by the model in [31]. Future work is also encouraged to establish the Pareto-optimal consistency–robustness trade-off in online portfolio selection. This work represents one operating point via RAM; whether the frontier can be improved (and whether RAM is Pareto-optimal) remains open.

## Acknowledgments

We thank the anonymous reviewers for their constructive feedback. Ziliang Zhang gratefully acknowledges the support of his wife, Wenyu Liu, and his parents, Chuanqing Zhang and Zhenying Huang.

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

# A   Technical Appendices and Supplementary Material

**Lemma A.1** (Fractional betting). *Let the forecast be wrong with independent probability $\epsilon \in (0, 1)$. For two assets whose return factors in period $i$ satisfy:*

$$(x_1(i), x_2(i)) = \begin{cases} (2, \frac{1}{2}), & \text{forecast is correct} \\ (\frac{1}{2}, 2), & \text{forecast is wrong} \end{cases}$$

*For any constant $\lambda \in [0, 1]$, let the portfolio be $\mathbf{b}(i) = \lambda \mathbf{e}_{\sigma(i)} + (1 - \lambda)(\frac{1}{2}, \frac{1}{2})$ where $\mathbf{e}_{\sigma(i)}$ puts weight $1$ on the predicted winner and $0$ on the other. The following holds:*

1. *Expected wealth.*

$$\mathbb{E}(S_n) = \left(\frac{5}{4}\right)^n \left(1 + \frac{3}{5}\lambda(1 - 2\epsilon)\right)^n$$

2. *Expected per-period log-growth.*

$$W_n(\lambda, \epsilon) := \frac{1}{n}\mathbb{E}\left[\log S_n\right] = \log \frac{5}{4} + (1 - \epsilon)\log\left(1 + \frac{3}{5}\lambda\right) + \epsilon \log\left(1 - \frac{3}{5}\lambda\right)$$

3. *Zero-growth error rate. For every fixed $\lambda \geq \frac{1}{3}$, there exists a unique*

$$\epsilon^*(\lambda) = \frac{\log \frac{5}{4} + \log\left(1 + \frac{3}{5}\lambda\right)}{\log\left(1 + \frac{3}{5}\lambda\right) - \log\left(1 - \frac{3}{5}\lambda\right)} \in \left(0, \frac{1}{2}\right)$$

   *such that $W_n = 0$. For $\epsilon > \epsilon^*(\lambda)$, $W_n < 0$.*

4. *Exponential competitive gap. Let $S_n^* = 2^n$ be the hindsight-optimum. Then*

$$\frac{S_n^*}{\mathbb{E}[S_n]} = \left(\frac{8}{5}\right)^n \left(1 + \frac{3}{5}\lambda(1 - 2\epsilon)\right)^{-n}$$

   *which grows like $e^{\Omega(n)}$ whenever $\lambda > 0$ and $\epsilon > 0$.*

*Proof.* (1) The per-period growth is $R(i) := \mathbf{b}(i) \cdot \mathbf{x}(i) = \frac{5}{4} \pm \frac{3}{4}\lambda$. When predictions are correct $R^+(i) = \frac{5}{4} + \frac{3}{4}\lambda$, when predictions are wrong $R^-(i) = \frac{5}{4} - \frac{3}{4}\lambda$. Since $R(i)$ are i.i.d,

$$\mathbb{E}[S_n] = \mathbb{E}\left[\prod_{i=1}^{n} R(i)\right] = \prod_{i=1}^{n} \mathbb{E}\left[R(i)\right] = (\mathbb{E}[R(1)])^n$$

$$= \left[(1 - \epsilon)\left(\frac{5}{4} + \frac{3}{4}\lambda\right) + \epsilon\left(\frac{5}{4} - \frac{3}{4}\lambda\right)\right]^n = \left[\frac{5}{4} + \frac{3}{4}\lambda - \frac{3}{2}\lambda\epsilon\right]^n$$

$$= \left[\frac{5}{4} + \frac{3}{4}\lambda(1 - 2\epsilon)\right]^n = \left(\frac{5}{4}\right)^n \left(1 + \frac{3}{5}\lambda(1 - 2\epsilon)\right)^n$$

(2) Because $R(i)$ are i.i.d.,

$$W_n(\lambda, \epsilon) = \frac{1}{n}\mathbb{E}[\log S_n] = \frac{1}{n}\sum_{i=1}^{n} \mathbb{E}[\log R(i)] = \frac{1}{n}\sum_{i=1}^{n} \mathbb{E}[\log R(1)] = \mathbb{E}[\log R(1)]$$

$$= (1 - \epsilon)\log \frac{5}{4}\left(1 + \frac{3}{5}\lambda\right) + \epsilon \log \frac{5}{4}\left(1 - \frac{3}{5}\lambda\right) = \log \frac{5}{4} + (1 - \epsilon)\log\left(1 + \frac{3}{5}\lambda\right) + \epsilon \log\left(1 - \frac{3}{5}\lambda\right)$$

(3) Observing $W_n(\lambda, \epsilon)$ is strictly decreasing on $\epsilon$, $\forall \lambda \in [0, 1]$:

$$\frac{dW_n(\lambda, \epsilon)}{d\epsilon} = \log\left(1 - \frac{3}{5}\lambda\right) - \log\left(1 + \frac{3}{5}\lambda\right) < 0$$

At $\epsilon = 0$, $W_n(\lambda, 0) = \log \frac{5}{4} + \log\left(1 + \frac{3}{5}\lambda\right) > 0$; At $\epsilon = \frac{1}{2}$, $W_n(\lambda, \frac{1}{2}) = \log \frac{5}{4} + \frac{1}{2}\log\left(1 - (\frac{3}{5}\lambda)^2\right)$, which is negative if and only if $1 - (\frac{3}{5}\lambda)^2 < (\frac{4}{5})^2 \Leftrightarrow \lambda > \frac{1}{3}$. If $\lambda > \frac{1}{3}$, $W_n(\lambda, \epsilon)$ is positive at $\epsilon = 0$,

negative at $\epsilon = \frac{1}{2}$ and strictly decreasing on $\epsilon$. With the intermediate value theorem, there exists exactly one root $\epsilon^* \in (0, \frac{1}{2})$ such that $W_n(\lambda, \epsilon^*) = 0$. Solving this yields:

$$\log \frac{5}{4} + \log \left(1 + \frac{3}{5}\lambda\right) - \epsilon^* \left[\log \left(1 + \frac{3}{5}\lambda\right) - \log \left(1 - \frac{3}{5}\lambda\right)\right] = 0$$

$$\epsilon^*(\lambda) = \frac{\log \frac{5}{4} + \log \left(1 + \frac{3}{5}\lambda\right)}{\log \left(1 + \frac{3}{5}\lambda\right) - \log \left(1 - \frac{3}{5}\lambda\right)} \in \left(0, \frac{1}{2}\right), \qquad \lambda \geq \frac{1}{3}$$

Because $W_n(\lambda, \epsilon)$ is strictly decreasing in $\epsilon$ and satisfies $W_n(\lambda, \epsilon^*(\lambda)) = 0$, it immediately follows that $W_n(\lambda, \epsilon) < 0, \forall \epsilon > \epsilon^*(\lambda)$.

(4)

$$\frac{S_n^*}{\mathbb{E}[S_n]} = \left(\frac{2}{\frac{5}{4} + \frac{3}{4}\lambda(1 - 2\epsilon)}\right)^n = \left(\frac{8}{5}\right)^n \left(1 + \frac{3}{5}\lambda(1 - 2\epsilon)\right)^{-n}$$

Since $1 + \frac{3}{5}\lambda(1 - 2\epsilon) < \frac{8}{5}$ whenever $\lambda > 0, \epsilon > 0$, the ratio grows exponentially in $n$.

$\square$

Table 5: Ticker compositions of three baskets on S&P 500.

| Basket | Constituent tickers |
|--------|---------------------|
| $\alpha$ | T, MSFT, NVDA, AMZN, V |
| $\beta$ | CSCO, MSFT, WRB, RF, T, NFLX, SBUX, BBY, ABT, BAX |
| $\gamma$ | CMCSA, AMP, HSIC, FSLR, FCX, DE, CE, VLO, BWA, PH, ANSS, AMZN, C, EXPE, FDX, TJX, WST, EMN, PGR, FAST, PODD, HST, ADM, NVDA, PAYX, BRO, MO, ESS, DTE, WEC |

