# OpenReview forum: "Online Portfolio Selection with ML Predictions"
_NeurIPS.cc/2025/Conference — NeurIPS 2025 poster_

### Official Review · Reviewer_Q1Qe · 2025-06-28

**Clarity:** 3
**Significance:** 2
**Originality:** 2
**Rating:** 3
**Confidence:** 4

**Summary:**

This paper studies the online portfolio optimization problem in a learning-augmented setting. The setting involves a trader who must manage, in an online manner, a portfolio of $m$ assets over a period of $n$ trading days; on each day, the current price of each asset is reveled. In the standard version of the problem, the trader has no access to any information about the future prices of assets. In the ML-augmented setting, each day a prediction is provided via an ordering of the anticipated profitability of the assets. This information may naturally be erroneous, and the objective is to obtain trading algorithms  that have good robustness (competitiveness against worst-case, adversarial predictions), consistency (performance assuming foolproof predictions) and smoothness (quantifiable degradation as a function of the prediction error).

The main theoretical result is an algorithm with the following guarantees: It has consistency at least $1/m$ times the hindsight optimal revenue; has robustness at least that of the best-known prediction-agnostic algorithms (informally, a profit equal to the geometric mean of all revealed prices) and quantifiable smoothness that interpolates between these two extremes. The upper bound is the only theoretical result, and no lower bounds (impossibility results) are given.

The work also gives experimental results using both synthetic data, in which predictions are generated according to a specified rule, as well as on more realistic setting in which the predictions come from a gradiant-boosting tree is trained on real data. The performance of the algorithm is then compared to a number of benchmark algorithms.

**Questions:**

1. Does your algorithm have connections to multiplicative-update approaches used in regret-based analysis of similar online problems?

2. Can you explain why all the algorithms you consider as benchmarks are quite old, are there no related recent algorithms in the literature that one could use?

3. What is the conjectured lower bound on the consistency? Also, could one potentially improve the consistency at the expense of robustness (or vice versa).

**Ethical Concerns:**

["NO or VERY MINOR ethics concerns only"]

**Final Justification:**

After the very helpful rebuttal by the authors, I am upgrading by overall score. I still believe that the paper is borderline for the following three  reasons:

*  The experimental evaluation is against very old algorithms, and seems to miss developments and baselines from regret-based optimization.

* The consistency is a fraction of $OPT$ only insofar as $m$ is treated as a constant, which may or mat not be a realistic assumption.

* Related to the above, an upper bound (impossibility result) would be very useful, if not necessary, in par with the recent progress in learning-augmented online algorithms. This does not necessarily imply the identification of the entire Pareto front between  consistency and robustness, but rather a few significant points, e.g., what if we require that the algorithm has robustness close to the "optimal" one. This becomes even more significant considering that the lower bound (main algorithm) comes from a very simple analysis, and the theoretical part of the paper is thus somewhat limited.

Notwithstanding the above observations, overall the work is interesting. I am not pushing for rejection, but I am not enthusiastic either.

**Limitations:**

I do not see any issues, other than explaining whether $m$ is assumed to be a constant or not.

**Paper Formatting Concerns:**

No concerns

**Quality:**

3

**Strengths And Weaknesses:**

Strengths:
* The problem is interesting and pertinent to the ML community. Using ML techniques to improve portfolio optimization is a problem with everyday applications.
*There are both theoretical and experimental results. The experimental analysis incorporates realistic prediction models.

Weaknesses:
* The main weakness is the claim that the consistency is "a constant factor from optimal". This is technically incorrect, unless the number of assets is considered constant. The statement in line 230-233 seems to suggest it is not. Regardless, the absence of a lower bound is critical. One is left with the overall impression that the analysis only scratches the surface of the problem, especially considering that the upper bound is technically not difficult to obtain.
* There are many related, albeit simpler problems such as one-max search and one-way trading that have been studied in several works under learning-augmented settings. I was looking forward to a discussion and comparison, especially considering that all these works present various types of impossibility results. On a similar vein, there are a lot of works on regret-based analysis of portfolio optimization under regret, but no discussion is given. I think the paper should also cite influential works and surveys by Li and Hoi, and Hazan.
* While the paper is well written overall, in several places, the discussion is very unclear due to the choice of language and terminology that obfuscates instead of clarifying, particularly in the introduction.
* As far as I can see, all algorithm benchmarks against which the obtain algorithm is compared come from the 90s. I find this extremely surprising given that portfolio optimization has been studied in many settings. The benchmark "ML" is very simple.

Overall evaluation, as summary of the above:

- Quality: Technically sound work, but does not address the full potential of possible solutions.
- Clarity: Overall the paper is well-written, but some parts use very obfuscating terminology, and misses important references.
- Significance: Well motivated an interesting problem, with an approach that is lacking lower bounds or some solid theoretical evidence.
- Originality: The obtained algorithm follows a simple idea with an analysis that is particularly involved, so the contribution is rather incremental.

---

> ### Author Rebuttal · Authors · 2025-07-30
>
> Thank you for your evaluation. We regret the notation and description imprecision that may have caused confusion and appreciate the chance to clarify. Below we respond to your comments in detail.
>
> **Clarification**
>
> Our submission introduces RAM, a learning-augmented portfolio strategy that updates asset weights directly from externally supplied ranking predictions. RAM is designed to provide consistency, robustness, and smoothness:
>
> - *Consistency* (good predictions with good performance): Theorem 2.1 Claim 2 shows that with perfect predictions, RAM achieves a payoff at least $\frac{1}{m}OPT$. RAM can in practice outperform this bound but can never fall below it under perfect forecasts.
> - *Robustness* (worst-case guarantee under adversarial predictions): Theorem 2.1 Claim 1 establishes that under adversarial predictions and arbitrary return sequences, RAM’s wealth is bounded below by the value line index (the geometric mean of each buy and holds). Because the arithmetic-geometric mean inequality is tight, this bound is essentially tight without introducing sequence-dependent variables.
> - *Smoothness* (error-sensitive degradation linked to prediction quality): Claim 1 also quantifies how RAM’s performance degrades as a function of prediction error, yielding a continuous transition between the robustness and consistency regimes.
>
> Both the robustness and consistency results are lower bounds; aside from the obvious upper bound OPT, no upper bounds are asserted. Finally, our paper contains no conjecture, and all stated results are rigorously proved and supported by empirical evaluation.
>
> **Is $m$ a constant?**
>
> We regret the earlier ambiguity. In our setting, the asset universe size $m$ is chosen once at the start of trading and remains fixed for all $n$ rounds. All theoretical results (including RAM's $\frac{1}{m}OPT$-consistency guarantee) are therefore stated with $m$ treated as a constant parameter. We recognize (as you highlighted) that $m$ is a priori a variable. The claim should be read as: the consistency is a constant factor from optimum under fixed $m$. We will clarify this in the revised manuscript to address the concern of the technical ambiguity.
>
> **Discussion to other problems and regret-based analysis**
>
> Thank you for pointing out the link between our portfolio-selection setting and the classical online problems of time series search (one-max search) and one-way trading. All three aim to maximize wealth against an adversary, yet they differ fundamentally in action space and reversibility:
>
> - Time series search: a single irreversible "stop" decision chooses one price; performance is gauged against the sequence maximum (or fluctuation ratio).
> - One-way trading: capital can flow from cash to an asset over multiple rounds, but never back; each partial conversion is irreversible.
> - Portfolio selection (our setting): at every round the investor freely rebalances across $m\geq 2$ assets, moving weight both into and out of any asset. Trades are thus fully reversible, and the benchmarks include: i) the best constant-rebalanced portfolio, and ii) the global optimum of all-in highest stake at each round.
>
> Consequently, one-way trading is a strict special case of portfolio selection with $m=2$ and a one-direction constraint, while time series search is the even narrower all-in-once variant. Our analysis therefore requires different techniques (e.g., continuous weight dynamics, the compounding effect of multiplication, logarithmic growth regret, etc.) and yields guarantees that do not follow directly from the simpler problems. Therefore, while one-max search and one-way trading are instructive pedagogical benchmarks, Cover's portfolio selection framework captures the full multi-asset landscape in which dynamic rebalancing lies at the core of modern online portfolio theory.
>
> We apologize for omitting this discussion. In the revision we will add a concise subsection contrasting the three settings and clarify why our focus is on the fully reversible, multi-asset scenario.
>
> Lines 25-45 and lines 98-108 of the introduction survey existing regret-based approaches, which typically establish asymptotic convergence (to the modest best constant-rebalanced portfolio benchmark) in logarithmic growth. By contrast, our analysis supplies non-asymptotic (exact) guarantees on realized wealth, which is the metric of greatest practical relevance for investors.
>
> **Additional references**
>
> We appreciate your pointer to the comprehensive survey “Online Portfolio Selection: A Survey”. We acknowledge its foundational importance and regret the omission. We will add an explicit citation in the introduction's literature review section to properly situate our contribution within existing methods.
>
> **Writing clarity**
>
> We agree that the manuscript’s exposition can be sharpened and have already enumerated specific revisions detailed in our reply to Reviewer jLHv, which we respectfully direct you to consult, as these updates will be integrated into the next version.
>
> You note that portions of the discussion (particularly in the introduction) are “very unclear”. To address this constructively, could you please identify the exact passages or arguments that require clarification? Concrete pointers will enable us to revise the text with the precision and transparency expected in the NeurIPS community.
>
> **Baselines**
>
> Thank you for raising the concern that some of our baselines appear dated. To the best of our knowledge, we are unaware of any existing algorithm that simultaneously (i) incorporates explicit external predictions into portfolio updates and (ii) provides formal guarantees of consistency, robustness, and error-smoothness. Hence, Cover’s side-information variant and its immediate extensions remain the most relevant points of comparison.
>
> Thank you for pointing that our ML component may appear overly simplistic. This choice is deliberate: to foreground the consistency-robustness trade-off we wish to study, we use a lightweight LightGBM forecaster that relies solely on lagged returns. The resulting signal is transparent, enabling us to sweep forecast quality without the confounding effects of a sophisticated model. This setup allows us to evaluate how RAM’s learning-augmented updates react across varying levels of predictive accuracy, rather than to demonstrate state-of-the-art forecasting performance.
>
> **Relationship between consistency and robustness**
>
> We appreciate the question “Can consistency be improved at the expense of robustness, or the reverse?”. In principle, yes. For example, RAM guarantees a consistency floor of $\frac{1}{m}OPT$ while remaining at least as robust as the value line. One could boost consistency by adopting an aggressive “all-in” strategy that invests solely in the asset with the highest predicted return each period. However, this approach suffers dramatic loss in robustness, rendering it of limited practical value. Determining the optimal consistency-robustness trade-off therefore remains an open problem. Theorem 2.1 quantifies a specific trade-off for RAM, but whether it is information-theoretically tight remains open. We will add a detailed discussion of these open directions in the conclusion of the revised manuscript.
>
> We hope these clarifications fully address your concerns. Thank you once again for the time and care you devoted to reviewing our work; your feedback has been invaluable.

---

> > ### Comment · Reviewer_Q1Qe · 2025-08-04
> >
> > Thank you very much for your detailed and helpful response. I would appreciate some further clarifications on the consistency. But first, allow me to clarify a few issues raised in the report. When I referred to ``lower bounds'' I meant impossibility results. I now see that my original choice of term was confusing, because a lower bound is a positive result since the problem is profit maximization. In regards to algorithms for other simpler problems (of which it is very well known that portfolio optimization is a generalization), at least two of them (time-series search and one-way trading) have been studied in learning-augmented settings, in several works. My point was that impossibility results established for these problems are of course pertinent to portfolio optimization, and this is the specific type of comparison that I believe should appear in the paper. The third point is about regret optimization. Again, I understand that your analysis is closer to competitive analysis, and this problem has not been studied under this framework because worst-case guarantees tend to be uninteresting or impractical for hard problems. But I believe there is a lot of recent work and efficient algorithms for regret analysis, so my question was why there was no such comparison in the *experimental* evaluation (not in the theoretical evaluation).
> >
> > With this in mind, I still cannot see from your response why $\frac{1}{m}OPT$ is a good lower bound as a function of $m$. I understand that you treat $m$ as constant, but in my opinion the asymptotics of the consistency are important, and an upper bound/ impossibility result would be very useful, if not necessary. You may want to clarify in case I missed something.
> >
> > I would also appreciate any further comments on my question 1 regarding multiplicative updates, unless you simply refer to related responses to other reviewers.

---

> > > ### Author Response · Authors · 2025-08-06
> > > **Rebuttal-Discussion**
> > >
> > > Thank you for your feedback.
> > >
> > > **Why is $\frac{1}{m}S^{OPT}$-consistency good**
> > >
> > > With fixed $m$, $\frac{1}{m}S^{OPT}$ is within a constant factor of the hindsight-optimal: $S^{OPT}=\prod_{i=1}^n\max_jx_j(i)$, which grows exponentially. Multiplying a positive wealth process by a positive constant leaves its multiplicative growth rate unchanged, so $\frac{1}{m}S^{OPT}$ and $S^{OPT}$ are growth-equivalent (the asymptotic of consistency to OPT). In logarithmic growth, the regret ablation to side information is:
> > >
> > > - $\frac{1}{n}\log S^{RAM}\geq\frac{1}{n}\log S^{OPT}-\frac{1}{n}\log m$
> > > - $\frac{1}{n}\log S^{UPSI}\geq\frac{1}{n}\log S^{OPT}-\frac{k(m-1)}{n}\log(n+1)$, where $k$ is the number of states.
> > >
> > > This shows RAM's consistency is much tighter. RAM delivers a strong trade-off guarantee than any existing portfolio algorithm (see essential reference at "The Cost of Achieving the Best Portfolio in Hindsight", the "impossibility result" of chasing BCRP).
> > >
> > > **Worst-case guarantee is indispensable**
> > >
> > > Worst-case analysis is far from "uninteresting". A portfolio that excels only under benign data can erase years of gains when predictions turn adversarial. Worst-case analysis offers a principled way to quantify this downside risk.
> > >
> > > Classic portfolio research benchmarks algorithms against BCRP to ensure asymptotic growth even in hostile return sequences. In learning-augmented settings the need is even sharper: predictions can amplify gains but, if mis-specified, can also induce catastrophic losses. RAM addresses this by maintaining above the value line, ensuring robustness against arbitrary prediction (and return) sequences while still leveraging accurate ones for superior growth.
> > >
> > > Thus worst-case guarantees provide the safety net every practitioner demands when deploying learning-enhanced strategies in real financial markets.
> > >
> > > **Impossibility result**
> > >
> > > By "impossibility results", we assume that you refer to the best possible consistency-robustness (C-R) tradeoff.
> > >
> > > A natural question is whether a portfolio can achieve consistency independent of $m$. This is attainable: an "all-in" strategy that fully invests in the highest predicted return at every step does achieve OPT consistency exactly. Yet this comes at a steep price: sacrificing robustness entirely. By contrast, **our guarantee of $\frac{1}{m}S^{OPT}$ may appear conservative when viewed in isolation, but it is paired with a robustness bound that matches the best prediction-agnostic portfolio algorithm.** That said, a rigorous competitive analysis of learning-augmented algorithms must substantiate the C-R trade-off across the entire spectrum, rather than evaluating at only one extreme (e.g., only consistency).
> > >
> > > Achieving the best C-R trade-off is an important long-term goal, but proving such optimality falls outside the scope of this work. Our work remains significant for two reasons:
> > >
> > > - Contribution. RAM is the first algorithm that quantitatively links portfolio performance to prediction quality (smoothness), providing rigorous guarantees of both consistency and robustness across the full spectrum of predictive accuracy. RAM represents a well-characterized point on the C-R frontier: our analysis demonstrates RAM simultaneously attains the stated consistency (matching OPT's growth up to a tighter bound) and the strongest known robustness among prediction-agnostic portfolio algorithms, underscoring its significance. We believe such analysis will stimulate further research at the intersection of mathematical finance and learning-augmented algorithms.
> > > - Precedent in learning-augmented algorithms. i) "Improving Online Algorithms with ML Predictions" (NeurIPS 2018) was one of the first to endow classical online problems (ski-rental and non-clairvoyant scheduling) with explicit C-R guarantees. Although it **did not** prove those guarantees optimal, the work was nonetheless accepted, has accrued hundreds of citations, and has since become the canonical framework adopted across caching, scheduling, sorting and other online settings. ii) Only years later did "Optimal Robustness-Consistency Trade-offs for Learning-Augmented Online Algorithms" (NeurIPS 2020) tighten the bounds in 2018, thereby **confirming that the absence of optimality proof in the 2018 paper in no way diminished its impact or publishability.**
> > >
> > > These examples illustrate a standard publication trajectory in our community: foundational paper that articulate clear, non-trivial trade-off guarantees are valuable even before tight optimality is settled. In the same spirit, our contribution establishes a rigorously justified point on the C-R frontier for portfolio selection, laying essential groundwork that future research can refine, just as later work sharpened the 2018 results.
> > >
> > > Finally, we apologize for overlooking the discussion of multiplicative-update algorithm. We address this in our response to Reviewer rUoK and will incorporate the regret-based logarithmic-growth analysis into the revised manuscript.

---

> > > > ### Comment · Reviewer_Q1Qe · 2025-08-07
> > > >
> > > > Thank you for your detailed response which I appreciate. Allow me a follow up comment / clarification. Concerning the consistency, my question was whether an upper bound as a function of $m$ is possible. Can you show that the best consistency is, say, at most $\frac{1}{\sqrt m}OPT$, or something similar? Are any upper bounds possible or rather hopeless for this problem? Do you have a conjecture about what is the right upper bound?
> > > >
> > > > Concerning "worst-case guarantees", I believe there was a misunderstanding. I meant that worst-case guarantees via competitive analysis, in the classic setting, may be very crude for some problems and may reflect only very extreme, hence "uninteresting" cases. I never intended this comment to be meant as "worst-case guarantees are uninteresting"; this would be flat-out wrong as you point out. As far as I know there are no non-trivial upper bounds for *standard* portfolio maximization, is this correct?
> > > >
> > > > Finally, concerning impossibility results, I did not refer explicitly to consistency/ robustness trade offs. I am in full agreement that the first step in keeping up with the worst-case competitive ratio while trying to optimize consistency is often a sufficient contribution on its own. I was wondering whether the upper bounds of previous works on related learning-augmented settings such as one-way trading have implications for the setting you study.

---

> > > > > ### Author Response · Authors · 2025-08-07
> > > > > **Rebuttal-Discussion**
> > > > >
> > > > > Thanks for your detailed response and clarification. We are certain that there are misunderstandings in our previous communication. We apologize for it and appreciate the opportunity to clarify.
> > > > >
> > > > > The very first point we want to clarify is that the upper bound/impossible result must be the so-called optimal consistency-robustness trade-offs, i.e., there should be no other interpretations. To see this, the (unconditional) upper bound for consistency is 1, achievable by “all-in the best”: it puts all the money into the asset with the highest return prediction at each round. Clearly, this algorithm achieves OPT under perfect predictions. Consistency is the competitive ratio under the perfect predictions and “all-in the best” achieves 1-consistency (and arbitrarily bad robustness). Note that it is common that an online learning-augmented algorithm can easily achieve the offline optimum under the perfect predictions, e.g., in portfolio construction and various scheduling problems. Therefore, the consistency is generally not considered by itself; it must be paired with a level of robustness. Consequently, the upper bound/impossibility result must be interpreted as the best possible consistency achievable for some fixed robustness, i.e., optimal consistency-robustness trade-offs.
> > > > >
> > > > > As we have discussed in the previous communication, this work does not consider the optimal consistency-robustness trade-offs; it presents a justified trade-off point—RAM. However, as 1) this is the very first algorithm with predictions study in this problem and 2) portfolio construction is a critical problem in mathematical finance, we believe this work will stimulate further research. We value the point raised by the reviewer that an optimal trade-off (i.e., impossibility result) is missing in this work; this, however, will be our future work. We would kindly request that the reviewer reconsider the significance of the contribution after this clarification.

---

### Official Review · Reviewer_jLHv · 2025-07-02

**Clarity:** 3
**Significance:** 4
**Originality:** 4
**Rating:** 5
**Confidence:** 5

**Summary:**

This paper proposes a new portfolio selection strategy that can incorporate any prediction of "ranking" of future returns, while maintaining the worst-case lower-bound guarantee.
The classical universal strategies represented by Cover's Universal Portfolio track the best constant strategy in hindsight, up to a polynomial factor that would diminish in the asymptotic regime. These classical approaches also enjoy the worst-case lower-bound guarantee.
However, since there exists high-quality predictions, it is highly desirable to incorporate such a predictive power in the portfolio selection.
The challenge is to achieve the best of both worlds, by incorporating predictions and guaranteeing not to go bankrupt.
The proposed strategy receives a prediction of "ranking" of returns (i.e., price relatives) for the next day, sort the betting according to the ranking, readjust the betting based on the actual returns, and repeat.
The computational complexity is also light, only requiring to sort $m$ values at each step.
The theoretical guarantee is nice, showing that they are guaranteed to be lower bounded by the value line index same as Cover's UP, but also only away by a constant fraction from the optimal strategy when the ranking oracle is perfect.
The experimental results demonstrate the practical relevance of the algorithm.

**Questions:**

## Suggestions on notation
There are too many inconsistent notation throughout. For example:
- For permutation:
  - Define $Q\_m$.
  - Given a permutation $\sigma(i)$ at day $i$, it seems that its evaluation at index $k$ is denoted by both $\sigma(i,k)$ and $\sigma(i)_k$. The prior notation is extremely confusing, so please use the latter, if I am not mistaken.
- After line 124, $x_{\sigma(i,k)}(i)\in \mathbf{x}(i)$ is not well-defined.
- It's super weird that daily error rate $\eta(i)$ is defined as such, and the total error rate is the multiplication of the "reciprocals". To my view, the daily error rate should be defined as $\eta(i) = \text{(gain with RAM)} / \text{(gain with perfect prediction)}$, which is the reciprocal of the current definition, so that the total error rate can be defined as the running multiplication of all daily error rates.
- Choose one terminology from {return, price relative}, or explicitly mention that you use them interchangeably.
- Choose a single notation for inner product. Currently both $\intercal$ and dot product notation are used.

## Typos/suggestions
- In line 130, would it be better to say "We adopt the geometric mean of buy-and-holds of each stock, which is referred to as the \emph{value line index}, as such a baseline. Formally, ..."?
- I appreciate that the authors try to put their contribution early in the introduction in line 134, but the execution is poor. The prediction error $\eta_n$ is neither defined nor referred to a future reference. The "rank-matching strategy" is also not defined at this point. Eq. (5) is not comprehensible. What is $j$ in $b_{(j)}$? This is a very bad notation.
- Line 165: though -> through
- In Eq. (7), := should be =.
- The statement in Eq.(8) in Theorem 2.1 (and so the expression after line 136) is very weird, as the second argument in $\max$ is another expression of $S\_n^{\rm RAM}$ as shown in Eq. (7). The first statement should be just the value line index as a lower bound, for the worst-case guarantee. If the authors want to keep it, maybe they can restate the alternative expression (in Eq. (7)) in Theorem 2.1 before the two items.
- After line 212, after the $\Longrightarrow$, consider having $1=\sum_{j=1}^m b_{(j)}(n+1)=...$.
- As alluded to earlier, the expressions after line 211 and line 214 are incomprehensible, whenever I see $b_k(...) \propto b_{(j)}(...)$.
- Table 2:
  - Comp. should be Comb. (as a shorthand for Combination)?
  - Consider using $p=...$ instead of the subscripts like 50% under RAM-R.
- In the experiment, consider plotting the log-wealth performance of RAM varying $p\in[1/2,1]$, while having some baseline performance indicated as horizontal lines. I believe that this will be very informative.

- Most importantly, I think the paper can largely benefit from revising Algorithm 1, and also revising the rest of the manuscript accordingly.
My biggest confusion was whether RAM requires $\mathbf{y}(i)$, which is the **true** price relative vector sorted by the prediction, or it just requires a ranking prediction, not the actual sorted relative vector. The confusion arises from the (too) early introduction of the quantity $\mathbf{y}(i)$ both in Problem Statement and Algorithm 1, which is only required for the sake of analysis. Please do not define it early on, and make clear that the algorithm simply requires ranking prediction. For Algorithm 1, my suggestion is to "time" each line from the algorithm's perspective as follows:
  - Remove line 3, as at this point the algorithm has no access to the price relative at day $i$.
  - In place of line 3, "Receive a ranking prediction $\hat{\sigma}(i)$ from oracle."
  - Consider removing $(i)$ in $\mathbf{b}^{\rm new}(i)$.
  - Line 7 should be $\mathbf{b}\_{\hat{\sigma}(i)\_j}^{\rm new} \gets \mathbf{b}\_j^\downarrow$
  - After line 8, add "Receive return / price relative $\mathbf{x}(i)$.
- And I believe that it is worth highlighting the key relation $(\mathbf{b}^{\rm new})^\intercal \mathbf{x}(i)= \mathbf{b}^\downarrow(i)^\intercal \mathbf{y}(i)$, maybe somewhere in the main text, but not in Algorithm 1.

I believe incorporating these will greatly improve the readability of the paper.

A final comment: I believe the proposed algorithm will undoubtedly stimulate further research within the mathematical finance community. I am also particularly excited to see whether this idea might influence recent developments in the application of universal portfolios to computational statistics (e.g., for anytime confidence intervals) and to online optimization (e.g., for parameter-free algorithms).

**Ethical Concerns:**

["NO or VERY MINOR ethics concerns only"]

**Final Justification:**

The authors' rebuttal sufficiently addressed my concerns on the clarity. I've updated my score from 3 to 5, increasing the score on Clarity from 1 to 3.

**Limitations:**

The authors clearly addressed the limitations in the last section.

**Quality:**

3

**Strengths And Weaknesses:**

## Strengths
The problem of portfolio selection with prediction is very interesting in theory and also practically important problem.
The proposed solution not only is very simple while naturally incorporating any ranking predictions, but also enjoys nice theoretical guarantees. To my understanding, the solution is new, original, and refreshing.

## Weaknesses
That said, I have a serious concern with the current form of writing, as the exposition can be much more improved.
After careful reading, I figured out what the actual algorithm is, but the current writing and poor notation makes it really hard to understand.
I think this paper has a VERY nice and important contribution, so it deserves better writing for its wider dissemination.

I will be happy to increase the score on Quality and Clarity if the authors promise to substantially revise the methodological part.
I listed a few suggestions below.

---

> ### Author Rebuttal · Authors · 2025-07-30
>
> Thank you for recognizing our contribution and for the detailed suggestions on improving the methodology section. Your feedback is invaluable, and we greatly appreciate the time and effort you invested in your review.
>
> We promise to substantially revising the methodological part to ensure greater clarity and accessibility, including: i) clarifying notations; ii) refining the algorithmic pseudocode; iii) improving descriptive text. As a first step, we would like to correct a typographical error in Theorem 2.1 (details below). We apologize for this oversight and will incorporate the correction, along all methodological improvements in the revised manuscript.
>
> **Typo (line 213)**
>
> The expression following $\geq$ should read
> $$\frac{1}{m}\prod_{i=1}^nx_{(1)}(i)$$
> not $x_{(1)}(j)$. Below we describe the planned changes in detail to address your concern:
>
> **Notations**
>
> - For permutations: i) We will add a descriptive sentence defining that $Q_m$ is the set of all possible permutations from $m$ assets; ii) Throughout, a permutation evaluated at position $k$ on day $i$ will be written $\sigma(i)_k$. We previously used $\sigma(i,k)$ to avoid nesting subscripts, but recognize this was imprecise and will adopt the clearer $\sigma(i)_k$ everywhere.
> - We will write $x_{(k)}(i)=x_{\sigma(i)_k}(i)\in\mathbf{x}(i)$, meaning the k-th largest return at time $i$, which is the component of the vector $\mathbf{x}(i)$ indexed by $\sigma(i)_k$.
> - We apologize for the confusion on the definition of error rate. When defining the daily error factor $\eta(i)$, we debated whether to use the ratio RAM/hindsight or its reciprocal. Intuitively, an error measure should decrease when performance improves and increase when it worsens. This interpretation favors hindsight/RAM, so that larger gains correspond to smaller error. However, this seems to complicate the expression and blur clarity. We will adopt your suggestions and use RAM/hindsight while explicitly explaining the meaning of “error” under this context.
> - On first use (Introduction, line 20) we will add the footnote: “Throughout, ‘return’ and ‘price relative’ are used interchangeably”.
> - We will use the dot product notation consistently in Algorithm 1 (line 9). The appearance of transpose will be replaced.
>
> **Typos/Suggestions**
>
> - We apologize for the oversight and will adopt your suggested correction on line 130.
> - i) We will add a forward reference for $\eta(i)$ and $\eta_n$ in the contribution section so readers can locate the definitions on them easily. ii) We will refine the description on “rank-matching strategy” and insert a forward reference where the mechanism is formally introduced. iii) We will state explicitly after equation (3) that $b_j$ denotes the $j$-th coordinate in a fixed basis, whereas $b_{(j)}$ is the $j$-th *largest* weight after the permutation applied by $\sigma(i)_j$. iv) The intended meaning for equation (5) is that larger weights are paired with higher predicted returns (one-to-one by rank). We will correct the notation accordingly.
> - We apologize for the oversight and will correct the typo on “though”.
> - In equation (7), you are correct: the equality should be "$=$" rather than "$:=$" as the error has been defined in equation (6). We will fix this.
> - We will incorporate your suggestion on equation (8): mention the alternative expression earlier and state explicitly that Claim 1 gives only the worst-case robustness lower bound.
> - We will add the transition proof after line 212 to improve logical flow.
> - The proofs after line 211 and 214 mark the switch from a fixed coordinate system to the permuted one. We will add explanatory text to clarify the change of basis.
> - For Tables: “Comp.” stands for composition (stock composition in Table 2; basket composition in Tables 3 and 4); we omit the parameter $p$ elsewhere to avoid redundancy but will reiterate this in the table.
> - Thank you for the suggestion on log-growth plots; we will include additional plots and discussion to illustrate smoothness more transparently.
> - Thank you for noting the ambiguity in the pseudocode. We will clarify that the algorithm only requires the predicted rank ordering; the numerical return estimates appear purely for theoretical exposition and can be removed from the implementation. In the revision we will: i) move the formal definition of the predictor from the problem statement to the analysis sections; and ii) update Algorithm 1 as suggested: line 3 will consume the ranking prediction, the portfolio update will use $\mathbf{b}^{new}$, the subscript in line 7 will be corrected, and an explicit "observe return" step will be added after line 8.
> - We will insert a brief clarification in the main-mechanism paragraph explicitly stating that $(\mathbf{b}^{new})^\top\mathbf{x}(i)=\mathbf{b}^{\downarrow}(i)^\top\mathbf{y}(i)$. Thank you for highlighting this point.
>
> We are sincerely grateful for your thorough and constructive review. Your detailed comments not only strengthened the methodological and expository aspects of our submission but also affirmed the broader relevance of our work to learning-augmented algorithms, mathematical finance, and machine learning research. We commit to a thorough revision for clarity, incorporating all of your recommendations so that the final manuscript is fully polished and ready for dissemination. Finally, we appreciate the time and expertise you devoted to guiding these improvements.

---

> ### Comment · Reviewer_jLHv · 2025-08-02
>
> I appreciate the authors' responses. With the points promised to be revised, I believe that this will become a solid contribution to the community. Adding discussions on some of the concerns raised by Reviewer Q1Qe will further improve the quality of the paper. I will update the score for acceptance!
>
> (Edit: fixed a typo.)

---

> > ### Author Response · Authors · 2025-08-03
> > **Rebuttal-Discussion**
> >
> > Thank you so much for your thoughtful review and steady support!
> >
> > Your insights and suggestions have genuinely shaped this work and will help move the field forward.

---

### Official Review · Reviewer_rUoK · 2025-07-03

**Clarity:** 3
**Significance:** 3
**Originality:** 3
**Rating:** 5
**Confidence:** 4

**Summary:**

This paper addresses the fundamental challenge in online portfolio selection of how to effectively use potentially noisy machine-learning (ML) predictions without sacrificing worst-case performance guarantees. The authors introduce a new learning-augmented algorithm called the Rebalanced Arithmetic Mean (RAM) portfolio. The core idea of RAM is to decouple the weight-generation process from the asset-selection process. It uses a robust, prediction-agnostic base strategy to generate weight magnitudes and then applies a "permutation layer," guided by an ML oracle's ranking of future returns, to assign these weights to specific assets.

The authors provide a rigorous theoretical analysis of RAM, proving that under arbitrary, even adversarial, market sequences, its performance has a strong lower bound tied to the market's geometric mean. At the same time, if the ML predictions are perfect, RAM's wealth is guaranteed to be at least a constant fraction (1/m) of the hindsight-optimal "all-in-best-asset" strategy. This formally captures the desired properties of a learning-augmented algorithm: consistency (good performance with good predictions) and robustness (a strong safety net with bad predictions). The paper substantiates these theoretical claims with comprehensive experiments on historical NYSE and S&P 500 data, using both synthetically-generated forecasts and a production-grade ML model.

**Questions:**

Questions for the Authors:

1. Your theoretical analysis guarantees that with perfect predictions, RAM achieves at least 1/m of the optimal wealth. In your experiments, especially on lower-dimensional portfolios, did you observe performance much closer to the optimum than this bound suggests? Is the 1/m factor a loose bound in practice?
2. The update rule for the weights in RAM is b(i+1) = (b_new(i) ⊙ x(i)) / R(i). This resembles the "follow the winner" dynamic of Exponential Gradient. Could you elaborate on the connection between RAM's update rule and other multiplicative-update algorithms?

3. In the S&P 500 experiments (Table 4), the standard ML approach of going all-in on the top-ranked asset sometimes performs very poorly (e.g., 0.532 in basket γ, regime A). In these cases, RAM still performs well. Does this suggest that the value of the ML oracle for RAM lies not just in identifying the single best asset, but in providing a reasonably good relative ordering across the board?

**Ethical Concerns:**

["NO or VERY MINOR ethics concerns only"]

**Limitations:**

No limitations

**Paper Formatting Concerns:**

OK

**Quality:**

3

**Strengths And Weaknesses:**

Strengths:

Elegant and Novel Algorithm Design: The core idea of RAM—decoupling weight generation from asset permutation—is both simple and powerful. It provides an intuitive and principled way to integrate external predictions into a classical, robust portfolio strategy. This two-stage architecture is a novel contribution to the learning-augmented algorithms literature.
Strong Theoretical Guarantees: The paper's main theoretical result (Theorem 2.1) is impressive. It formalizes the algorithm's consistency and robustness, showing that RAM smoothly interpolates between near-optimal performance with perfect predictions and a solid worst-case guarantee that matches other universal strategies. Proving these bounds for arbitrary (adversarial) return sequences is a significant technical achievement.
Comprehensive Empirical Validation: The experimental section is thorough and convincing. The authors test RAM under multiple scenarios:
A prediction-free baseline (RAM-R) to show the strength of the core rebalancing rule.
A controlled-noise study with synthetic predictions to demonstrate the smooth trade-off between prediction quality and performance.
A realistic deployment with a production-grade ML model (LightGBM) on recent, volatile market data (including the COVID-19 crash). This multi-faceted evaluation provides strong evidence for the practical utility of RAM.
Clarity and Motivation: The paper is very well-written. The introduction clearly frames the central tension between conservative universal portfolios and aggressive heuristic methods, motivating the need for a learning-augmented approach. The problem statement and algorithm are described clearly.

Weaknesses and Areas for Improvement:
Comparison to a Broader Range of Baselines: While the comparison to Universal Portfolio (UP) and Exponential Gradient (EG) is appropriate, the paper could be strengthened by including other well-known online portfolio strategies, such as mean-reversion (e.g., PAMR ) or pattern-matching algorithms, in the empirical evaluation. This would provide a more complete picture of where RAM stands in the landscape of existing methods.

Impact of Transaction Costs: The authors rightly mention in the conclusion that transaction costs are a key practical dimension that is not explored. While it's standard to omit this in initial theoretical work, the rebalancing nature of RAM suggests that transaction costs could be a significant factor. A brief discussion or even a simple simulation showing the impact of a fixed transaction cost would add significant practical depth.

Nature of the ML Oracle: The paper treats the ML model as a black-box ranker. While this is a valid and general approach, a little more detail on the features used for the LightGBM model in the S&P 500 experiments (beyond "contemporaneous and three lagged returns") could be beneficial for reproducibility and understanding.

---

> ### Author Rebuttal · Authors · 2025-07-30
>
> Thank you for recognizing the significance of our contribution and for your thoughtful evaluation. Your feedback is invaluable to the communities of learning-augmented algorithms, mathematical finance, and machine learning, and we appreciate the time and expertise you have devoted to this review. We address your concerns below.
>
> **Additional baselines**
>
> We appreciate the suggestion and agree that adding more algorithms could be informative. However, our study focuses on a narrower yet fundamental question: How does algorithmic performance change as the quality of an external prediction signal varies? To address this, our experimental design requires baselines that (i) explicitly accept the same prediction signal as input and (ii) possess a principled mechanism for exploiting or defending against that signal. Universal portfolio variants equipped with side information (UPSI, EGSI) satisfy (almost) both criteria, making them the most appropriate comparators:
>
> - Scope and assumptions: PAMR and other mean-reversion or pattern-matching strategies rely on specific market assumptions and do not consume predictions. Their objective and information sets therefore differ materially from RAM’s.
> - Practical Fairness: In our experiments we feed an identical (almost) prediction signal to RAM and UPSI/EGSI. This choice is conservative for RAM because RAM must remain robust even when the signal is adversarial, whereas UPSI/EGSI can still partially exploit it. For algorithms that ignore predictions altogether (mean reversion, pattern matching, etc.), any attempt at a “fair” comparison would either give them a free pass on adverse signals or bias the evaluation against prediction-aware methods.
>
> For these reasons, we limit the baseline set to UPSI/EGSI. We will add clarifying text to the manuscripts and an appendix discussion noting that incrementing non-prediction methods is non-trivial and left to future work.
>
> **Transaction cost**
>
> Transaction costs are crucial in practice. We originally experimented with incorporating them into RAM, but the resulting model proved sufficiently intricate that a full treatment would have overshadowed the main exposition. We therefore flagged the omission as a limitation in the conclusion. Below we summarize the insights we obtained and will incorporate as a brief discussion in our upcoming manuscript. Assume a fixed fee applies only when selling and let us analyze two rebalancing policies:
>
> - Full Liquidation: Every period the algorithm sells its entire holding and buys back according to the new weights. If the fee is $0.4%$%, wealth evolve as $S^{RAM,fee}_n=0.996^n\cdot S^{RAM,free}_n$. Thus, the drag compounds exponentially, and the cost drives the portfolio below its value line in the worst case. On the other hand, whether this cost drives consistency below $\frac{1}{m}OPT$ remains open.
> - Optimal partial rebalancer: Instead of liquidating everything, the algorithm sells only the excess of assets forecast to underperform. This greatly reduces turnover, but the net impact depends on the realized return sequence. While intuitively superior to full liquidation, it complicates the theory and can still deteriorate robustness. It remains open whether bounds analogous to the original ones survive up to a constant, logarithmic, or polynomial factor.
>
> Furthermore, extending RAM to cost-aware rebalancing requires a new adversarial model and proof technique, work we believe merits a separate study. Including it here would significantly enlarge the paper and blur the main contribution. We will add the above discussion to the revised manuscript and highlight a cost-aware extension of RAM as an important direction for future research.
>
> **Reproducibility**
>
> Thank you for highlighting the reproducibility concern, and we apologize for the oversight. We summarize the key input features employed by our LightGBM model from line 297 to 302. To keep the main text centered on the algorithmic contribution, the complete hyper-parameter table together with the training and evaluation scripts are relocated to the supplementary material (line 302-304).
>
> To enhance reproducibility in our upcoming manuscript, we will cite a GitHub link in the abstract containing the full training code and the exact LightGBM parameter grid (as well as all codes needed for replicating our experimental results). We hope these additions can address the reproducibility concern.
>
> **Tightness on consistency**
>
> Thank you for raising this point. On real data, the bound becomes tight very quickly because wealth compounds exponentially. i) For a two-asset illustration, the inequality on line 213 telescopes to $S^{RAM}_n=(S^{OPT}_n+S^{ADV}_n)/2$, where ADV denotes the worst-performing asset each round. Over even moderate horizons, $S^{ADV}_n$ becomes negligible when compared to $S^{OPT}_n$, leaving $S^{RAM}_n\approx\frac{1}{2}S^{OPT}_n$. Hence the realized ratio approaches the theoretical floor in practice. ii) A similar argument extends to high-dimensional portfolios: the product of the period-wise "minimum" (any return except the highest one) decays so rapidly when compared to OPT that the empirical ratio is close to $\frac{1}{m}$.
>
> We will add a concise version of the above insights in our empirical section to the revised manuscript for clarity.
>
> **Relation to multiplicative update algorithms**
>
> Thank you for highlighting the connection. RAM does share the multiplicative flavor of follow-the-winner algorithms such as EG, but its design philosophy diverges in three important ways:
>
> - Different objectives. EG balances two forces: it rewards assets that performed well in the most recent period and penalizes larger deviations using relative entropy. This built-in conservatism smooths the portfolio path. By contrast, RAM is prediction-centric: at every round, it re-weights fully toward the ordering supplied by the external predictor, with no explicit regularizer. If predictor says asset X will outperform, RAM immediately assigns it the highest weight, regardless of yesterday’s allocation.
> - Distinct risk profile. Because RAM can flip a large fraction of its wealth onto a single predicted winner, one bad prediction can sharply reduce its value. We accept this vulnerability intentionally; the algorithm is analyzed in two regimes: consistent with good predictions and robust with bad predictions. EG lacks this dual guarantee: it is more stable but sacrifices (the extent depends on the learning rate) the upside that RAM captures when the predictor is informative.
> - Theoretical guarantees. EG is competitive with BCRP (or BSCRP if side information is provided) in the asymptotic regime. RAM instead interpolates between the robustness and consistency extremes, because it is willing to follow the predictor without the inertia that EG enforces.
>
> **Significance on relative ordering**
>
> Thank you for pointing this out. RAM transforms the entire predicted order into a graded weight vector, assigning higher-ranked assets with more weights while retaining positive shares for lower-ranked ones. Thus, even if the oracle misses the single best shot, any residual accuracy in the rest of the ranking directs capital toward several assets that ultimately do well (under the context of reasonably good relative ordering). The ML all-in strategy by contrast, has no such hedge. Theorem 2.1 bounds RAM’s loss when predictions are adversarial (above value line index). This guarantee is what protects the portfolio in the worst-case top-rank flip you highlight. However, while the whole order matters, the first rank is still the single largest weight (at least $\frac{1}{m}$). A wrong call there reduces growth more than a low-rank error from any other “good” assets. To answer your question, a reasonably accurate ranking across the board does enhance RAM’s performance, but the size of the gain depends on the specific (and potentially arbitrary) return path. Because such paths can vary arbitrarily, it is difficult to provide a tighter, sequence-independent analysis beyond the universal bound (smoothness) in Theorem 2.1.
>
> Thank you again for your valuable feedback. We hope the above response fully addresses your concerns. We will incorporate these clarifications into the revised manuscript and appreciate your continued support.

---

> ### Author Response · Authors · 2025-08-09
> **Rebuttal-Discussion**
>
> Dear Reviewer,
>
> We would like to kindly remind that the discussion period will close soon. If you would like any additional justification or have any concerns about our responses or the manuscript, please let us know and we will address them promptly.
>
> Thank you for your time, and we apologize for the interruption.

---

### Note · Authors · 2025-08-12

We thank the reviewers for their constructive feedback and appreciate their recognition and support of this paper. They have substantially strengthened the paper. In this final remark, we would like to provide summaries most relevant to the Area Chair's decision. Our goal is to close the loop on remaining points and clarify the paper's contributions.

We have addressed the reviewers' concerns throughout the rebuttal. For Reviewer jLHv, we clarified methodological ambiguities and will incorporate significant revisions to improve clarity and reproducibility. For Reviewer Q1Qe, we have addressed all the concerns. In particular, in the discussion about the paper missing the "impossibility results", we acknowledge that such an analysis falls within the optimal consistency-robustness trade-offs study, which goes beyond the scope of this work. We acknowledge that this should be an immediate future work following this paper, and we shall clarify this in the paper.

We would like to reinforce that the key contributions of the paper are 1) the first study of algorithms with ML predictions in portfolio selection and 2) the first algorithm RAM with robustness (under adversarial predictions), consistency (with accurate predictions), and smoothness (graceful degradation with prediction error) under arbitrary return sequences. Online portfolio selection is a core problem in mathematical finance. As black-box ML predictions become pervasive, it is pressing and timely to consider leveraging their predictive power while preserving the worst-case protections when predictions fail, especially in mission-critical settings like investment. This work pushes the boundary towards trustworthy ML applications for online optimization. We expect this research to stimulate many subsequent follow-up works to boost the fields of algorithms with predictions, ML, and mathematical finance.

Thank you for considering our submission.

---

### Decision · Program_Chairs · 2025-09-17

**Decision:**

Accept (poster)

**Comment:**

The paper proposes an online portfolio selection algorithm which leverages ML predictions. The proposed algorithm guarantees a worst-case return when the ML predictions are completely wrong, while yielding higher wealth when the predictions are accurate. The authors also derive a prediction error-dependent lower bound on the resulting wealth.

Reviewers overall agree that the paper makes a significant contribution by proposing an algorithm that leverages ML predictions in online portfolio selection in a safe way, while guaranteeing high returns when the predictions are correct. One main concern is the clarity of writing. Authors are strongly encouraged to incorporate reviewers’ suggestions to improve the clarity.